# The Usability of ICTs in People with Visual Disabilities: A Challenge in Spain

**DOI:** 10.3390/ijerph191710782

**Published:** 2022-08-30

**Authors:** Fiorella Fuentes, Antonia Moreno, Fernando Díez

**Affiliations:** 1Universitat Oberta de Catalunya (UOC), Av. Tibidabo, 39-43, 08035 Barcelona, Spain; 2University of Deusto, Avenida de las Universidades, 24, 48007 Bilbao, Spain

**Keywords:** accessibility, tiflotechnology, technical aids, literacy, low vision

## Abstract

The use of ICTs provides autonomy, equity, and social inclusion to people with visual disabilities. The National Organization of the Spanish Blind (ONCE) offers its 70,462 legally-blind people the necessary resources for the usability of ICTs. Still, most individuals with visual disabilities do not have a similar support system. This research aims to expose and make visible the importance and need for ICTs usage in this group. The qualitative approach has allowed the modeling of a contextualized inductive process through two heterogeneous discussion groups: eight individuals with legal blindness and six with moderate visual impairment, as well as three in-depth interviews with experts in typhlotechnology, accessibility, and low vision. The following has been verified: there is a lot of misinformation among people with visual disabilities outside the coverage of ONCE; accessibility is still not a priority for companies and institutions when creating and developing products and services with Design for All; the need for more professionals to advise and train users with blindness and low vision is clear. In Spain, there are almost a million visually-impaired people not affiliated with ONCE, for whom access to technical aids and digital literacy is a priority problem in which the Government should intervene.

## 1. Introduction

The WHO defines eHealth as the use of information and communication technologies (ICTs) for health, while mHealth is defined as medical practice and public health supported by mobile devices. Various studies have shown that eHealth (or ICTs in health) increases health services’ quality, efficiency, and coverage [1].

The leading role of ICTs in the daily life of the global population is unquestionable. However, society’s commitment is still timid when developing products based on Universal Design criteria. Thus, sensitization and awareness are essential to ensure that society works towards a universal development that favors both the global population and the group of individuals with visual disabilities that we will focus on in this research.

Vision is the most dominant of the senses, playing a fundamental role in all aspects of life. When we talk about visual impairment, we refer to the impairments, limitations, and restrictions that a person with an eye disease faces when interacting with their physical, social, or attitudinal environment [2].

Moderate visual impairment (visual acuity less than 6/18 or equal to or greater than 6/60) and severe visual impairment (visual acuity less than 6/60 or equal to or greater than 3/60) are grouped under the term “low vision”. Typically, low vision and blindness jointly represent the total number of cases of visual impairment.

At least 2.2 billion people worldwide have impaired near or distant vision. In addition, population growth and aging are projected to increase the risk of more people being affected by vision impairment in the world [3].

According to the Survey on Disability, Personal Autonomy and Dependency Situations—EDAD published in 2022 [4]—there are 4.38 million people with disabilities in Spain. Of these, 23.6% of the total, that is, 1,051,300 have visual disabilities.

People with visual disabilities have, in the knowledge society, the opportunity to access information that was previously conditioned. Universal access to information means everyone has the right to seek, receive and impart information. This right is an integral part of the right to freedom of expression [5].

In Spain, people with disabilities were not visible in society, with few exceptions, since they did not have the technical means, support, or necessary personnel. This is why people with hearing and visual disabilities were the first to unite to defend common rights. Thus, in 1938 different groups of visually impaired people from various regions of Spain came together to form the National Organization of the Spanish Blind, hereinafter ONCE [6].

This organization has created a model of social provision in Spain that is unique in the world for people with blindness and visual impairment. The conditions a person must meet to join are: having Spanish nationality and having at least one of the following visual requirements in both eyes: visual acuity equal to or less than 0.1 obtained with the best possible optical correction and visual field reduced to 10 degrees or less. There are currently more than 70,462 people (children and elderly) with recognized legal blindness who are affiliated with ONCE, where they are encouraged to achieve their personal autonomy with personalized, specialized, and specific attention [7,8].

Therefore, this research aims to expose and make visible the reality in Spain inside and outside ONCE and bring awareness to the importance and need for the usability of ICTs and digital literacy in people with visual disabilities. This way, the goal is to provide them with greater inclusion, equality, and participation in society. In short, we intend to improve the quality of life of these people and implement health intervention systems.

### 1.1. Digital Literacy As a Necessary Process for Improving Quality of Life

In the information and knowledge society, digital literacy has great force. It has been discussed since the early 1990s and is also known as digital information literacy. Gilster and Pool [9] define it as the set of socio-cognitive skills through which the process of transforming information into knowledge can be selected, processed, analyzed, and reported.

Thus, in addition to learning to seek and transmit information and knowledge, build and disseminate audiovisual messages, to train people so that they can intervene and develop in the new virtual environments and the creative use of human and material resources, it is essential to assume responsibility for education in the information society, as this is the way that contributes to reducing digital inequality [10].

Information and communication technologies can improve the quality of life of blind or partially sighted people. Some of these technologies based purely on the development of specific software are not patentable and must be protected through proper management of copyright associated with intellectual property [11].

People with visual disabilities can access content on the network as long as they have a typhlotechnical literacy [12] or specific training in the use of technical and typhlotechnical aids, according to the degree of visual impairment and the individual needs of each person.

Typhlotechnical literacy allows users with blindness and low vision to access a word processor, surf the net, access online training or remote work, use, in many cases, only the key commands and the visual capacity that is had or not, navigate looking for information, access a virtual campus of education, buy via e-commerce, access an intranet, and carry out a public administration procedure or interact in a social network, which, in the case of a phone or tablet, will be converted to finger gestures.

In order to achieve this skill, it is recommended to have typhlotechnology instructors [13]. Still, this will be given in one way or another depending on the individual conditions of each person.

Likewise, computer literacy [14] is also necessary as the knowledge of how to access the computer (folders, desktop, email), use a word processor, or any computer program will be achieved through the aids techniques or typhlotechnics. Thus, in the initiation phase, such literacy is essential.

Digital, computer, and typhotechnical literacy become crucial for people with visual disabilities, as, unlike other global users, self-learning might not be possible in their case at an initiation point. The impossibility to learn by themselves has caused digital and typhotechnical illiteracy and, in turn, brought inequality, exclusion, and inhibition.

### 1.2. Access to ICTs by Visually-Impaired People

The personal development of individuals with visual disabilities depends mainly on their qualifications in ICTs management. Accessibility [15] is essential and necessary, as well as technical aids and typhlotechnology as support resources for its usability.

For all this, correct digital, computer, and typhotechnical literacy are essential because a satisfactory web-browsing experience would not be achieved without them. This literacy also provides the use of the innumerable possibilities offered by computer programs and mobile applications [16] to be able to access and contribute to the information and knowledge to which the entire population is entitled.

There is no unification of criteria when defining what accessibility is. Some authors differentiate between the concept of accessibility and that of Universal Design and/or Design for All. Accessibility in Design for All refers to the specifications designed for people with disabilities, and Universal Design refers to access for the entire population without making differences between specific needs. The latter refers to human diversity in general [6].

According to the Web Accessibility International (WAI) guidelines, widely considered the international standard of the World Wide Web Consortium (W3C), Web Accessibility means that websites, tools, and technologies are designed and developed in such a way that people with disabilities can use them. More specifically, people can perceive the Web, understand it, navigate it, interact with it and contribute to it.

Web Accessibility encompasses all disabilities (hearing, cognitive, neurological, physical, speech, and visual) that affect access to the Web. It is important to emphasize that making mobile applications and web pages more accessible implies a direct transfer to society, which favors people with not only disabilities but also any other user, as it has been shown that accessibility benefits everyone [17].

Accessibility is a problem treated and widely considered both nationally and internationally. For this reason, the Accessibility Standards of the Portal of the Electronic Public Administration of the Government of Spain [18] are mentioned, from which some of the resources are collected that allow specifying the characteristics that must be met in the contents available through Internet Web technologies, Intranets and other types of computer networks, so that they can be used, autonomously or through the relevant technical aids, by most people, including people with disabilities and the elderly.

The UNE-EN 301 549:2022 standard is the Spanish version of the European standard EN 301 549 V3.2.1 (2021-03) “Accessibility requirements for ICT products and services applicable to public procurement in Europe”. Like its predecessor, it establishes the functional requirements to guarantee that ICT products and services are accessible to everyone, for example, from mobile phones to computers through websites. Thus, the accessibility guidelines, the Web Content Accessibility Guidelines (WCAG) 2.1, include the latest version of the “international recommendations” on how to make Web content accessible from the W3C (World Wide Web Consortium) to people with disabilities. Today, these are just some recommendations, but in the Spanish case, there is the Royal Decree 1112/2018, which does oblige developers and evaluators of mobile applications and web pages to comply with the previous harmonized standard of the Accessibility Directive: EN 301549 V2.1.2:2018, Government of Spain [18].

However, there are still companies that, when developing their websites, mobile applications, or intranet, find that accessibility is a matter of CSR policies or simply of an image since there is no international legislation that indicates the accessibility of the website or mobile application is mandatory. It is thus not about designing them for people with disabilities but, in general, for any user who cannot browse said website for whatever reason.

When a web page or mobile application is created from the outset with Universal Design or Design for All criteria, it is accessible. Although it may present some errors in the interface, they can easily be solved. However, when a web platform has been created without any accessibility criteria, it can show many errors in the interface that, when it comes to solving it, will mean more than one inconvenience, causing in many cases its redesign, with the consequent economic costs.

### 1.3. Technical Aids and Typhlotechnology

Technical aids, also known as assistive products, refer to any product (including devices, equipment, instruments, and software), specially manufactured or commercially available, used by or for people with disabilities. They are intended to facilitate participation, protect, support, train, measure, replace bodily functions/structures and activities, or prevent impairments, activity limitations, or participation restrictions [19].

The term typhlotechnology, which comes from the Greek Tiflo (blind), was incorporated into the Dictionary of the Royal Academy of the Spanish Language in 2008, where it is defined as the “study of the adaptation of procedures and techniques for use by the blind” [20]. This is how typhlotechnology is known as the set of techniques, knowledge, and resources aimed at providing the blind and visually impaired with the appropriate means for the correct use of technology to favor their autonomy and full social, labor, and educational integration [21].

Not all technological devices can be used by people who are blind or have low vision. For this reason, technical and typhlotechnical aids emerge as the ideal adaptation so that people with visual disabilities can use these devices and thus avoid social exclusion and reduce the digital divide.

It is important to add that smartphones are one of the devices that have evolved the most in recent years. Thanks to their portability, they are true pocket computers that allow us to use many applications anywhere, at any time. With them, the concept has been extended to the so-called mobile typhlotechnology [22]. Thus, many applications have been appearing within mHealth, transforming them into an invaluable and versatile assistance tool that allows users with visual disabilities to be more self-sufficient [23].

This is how the standard applications that serve as technical aids for these users arise. In the case of smartphones, simply by acquiring a terminal, they can already, after the corresponding configuration, access said terminal and autonomously download the necessary applications.

There are different technical aids divided into devices (physical instruments), programs (computer programs), and applications (mobile applications).

Among the different physical instruments that we find in the market is the Braille Line. It is a tool that allows communication between the computer and the user by transcribing the texts on the screen into braille, as long as they are in an accessible format for the communication software. It comprises cells made up of eight mobile stems, which correspond to the formation in relief of the characters in computerized braille. In addition, there are advanced braille lines and keyboards on the market, which can be connected via BlueTooth and USB to a mobile phone, a PDA, or a computer [24].

The electronic magnification systems for information access are electronic character-magnification systems with autofocus and working modes in natural, artificial, positive, and negative colors. There are usually two electronic magnifying devices, one large for home use, known as a television magnifier, and another pocket-sized one to carry with oneself on the move, called an electronic magnifier. It is another option for access to information for people with severe and moderate visual impairment [25].

The screen reader is a program that allows the screen’s content to be recognized and reproduced using voice synthesis. Generally, they are used by people with blindness or severe visual impairment [24].

The screen magnifier is another program designed to allow people with low vision to access reading on a computer or mobile device, and the programs it contains thanks to the features they offer to customize the size, shape, colors, mouse pointer, etc. In addition to enlarging the characters, they read the running programs and documents [25].

Concerning mobile applications, the screen reader is included in the set of smartphone accessibility services, both on Android and IOS. It is installed by default on most smartphones and tablets. It provides voice feedback that reads the content on the screen while offering spoken messages and notifications.

Likewise, as Muñoz [26] points out, grayscale options, designed for people with color differentiation problems, can convert all their interfaces to grayscale. The reverse contrast option changes screen colors from dark on light to light on dark. With customizable fonts, it increases the size and style of the font displayed on the device. Moreover, the zoom-in browser option allows you to increase the size of text and images in web content.

Voice synthesizers, screen magnifiers, braille lines, mobile applications, screen readers, GPS, color recognition, audio description, and others are a small sample of the range of technological aids for people with visual disabilities. They all provide the opportunity to access ICTs, information and knowledge, training, the labor market, citizen participation, social relations, and much more.

## 2. Materials and Methods

For the present study, we summoned men and women between the ages of 18 and 60 with visual disabilities residing in Spain, where ONCE exists as an organization with a differentiated model compared to other countries. In order to know their experiences with using ICTs first-hand, two discussion groups have been formed in which affiliated and non-affiliated participants of this organization have participated.

On the other hand, two experts in accessibility and typhlotechnology from ONCE and an expert in low vision have been interviewed to share their experience and points of view concerning information technologies within our analysis group.

In order to achieve the delimited objective, it has been decided to use a qualitative approach since it is beneficial in deepening the expectations, knowledge, experiences, and needs of people with visual disabilities concerning the usability of ICTs. This approach [27] models an inductive process contextualized in a natural environment because, in the data collection, a close relationship is established between the research participants, subtracting their experiences and ideologies to the detriment of the use of a default measuring instrument.

With the qualitative approach, a wide range of ideas and interpretations have been obtained that enrich the purpose of the investigation. The final scope of the qualitative study is to understand a complex social phenomenon, such as the one that concerns us. That is why we are using [28] all kinds of sources and materials that help us describe the routine and meaning of problematic situations in people’s lives, which help evaluate the perspectives of the people involved in the problem we are studying.

The techniques used for data collection are discussion groups and in-depth interviews, based on grounded theory as a method that involves the simultaneous collection and analysis of data. This method is characterized by being flexible since different research techniques can be combined, whose analysis and contrast allow methodical triangulation, which is appropriate to address the object of study and generate a theory about it [29].

### 2.1. Scope of Study

It is necessary to point out that Spain, unlike other countries in the world, has the figure of the ONCE organization. It is a non-profit organization of general interest recognized by Law 5/29 March 2011, on Social Economy, as a public law corporation, by and for the blind, and whose specific regulations confer the consideration of a singular entity of the social economy. ONCE maintains its legal personality and total capacity to act, as well as the self-organization to carry out its activities with leaders democratically elected by the members of the organization [30].

Men and women between the ages of 18 and 60 with visual disabilities who are affiliated and not affiliated with ONCE have been invited to participate in this research. The study has been limited to this age range since the stage of school education before 18 years of age, with special requirements, does have assistance in case of need. On the other hand, the age has been limited to 60 years because, after that age, additional circumstances begin to appear that can blur the point of focus that has been traced.

On the one hand, said sample comprises users affiliated with the ONCE who have recognized legal blindness (acuity less than 0.1 or visual field less than 10°). Making visible the experience of using ICTs is a particular case for the ONCE members in Spain as it is a source of opportunities in terms of advice, training, and means of support to achieve their autonomy, and that does not exist in other countries.

On the other hand, the sample comprises Spanish residents with moderate visual impairment, that is, who have a visual field of less than 20°. With them, we will make visible their experience regarding the use of ICTs from outside ONCE but as members of specific associations for people with visual disabilities. These non-profit associations, in addition to raising awareness of pathologies that affect vision, are a meeting point for people with visual disabilities and provide a support network with other members with similar characteristics in terms of their disability.

### 2.2. In-Depth Interviews

It has been decided to carry out in-depth semi-directed interviews, as it is an appropriate technique to delve into our object of study, which is the usability of ICTs by people with visual disabilities. This technique is effective for obtaining relevant data and finding facts, phenomena, or social situations. Knowing the approach of each of the experts who have been interviewed individually has made it possible to combine the scope of their perceptions, concerns, and experience with the ICTs by people with visual disabilities in Spain and thus, better understand the situation of this group.

This in-depth interview’s creative process of exhaustive inquiry consists of applying the qualitative technique with a high adaptation and invention to each new situation. This innovative process and intense personal interaction enable a higher level of persuasion and harmony with the interviewee, giving rise to a relationship based on a climate of trust [31].

Three experts have been summoned, each with a respective specialization, presented below:

#### 2.2.1. Expert 1

Typhlotechnology and Braille Instructor (ITB) in the Territorial Delegation of ONCE in Aragon. He is a Social Graduate and has studied computer science courses and a multitude of typhlotechnology studies at ONCE. He has worked for 36 years in this organization.

#### 2.2.2. Expert 2

Head of the Information and Accessibility Unit of the ONCE general management since 2019. He has computer studies and a Master’s degree in accessibility.

#### 2.2.3. Expert 3

He has been dedicated to low vision consulting, development, and research since 2005. He studied Telecommunications Engineering and has a Master’s Degree in Universal Accessibility and another in Education and Communication on the Internet.

### 2.3. Discussion Groups

With this qualitative technique, more information has been obtained regarding the perceptions, ideas, attitudes, or experiences of people with visual disabilities and their use of ICTs. Discussion groups are a form of unstructured interview with the purpose of capturing qualitative data through discussions. They resort to speech and are characterized by their great flexibility, as well as in-depth interviews [31].

In this section, we would like to recall that visual impairment includes moderate visual impairment and severe visual impairment. In moderate visual impairment, the corrected distance visual acuity is between 0.3 and 0.1, or the visual field is less than 20%. In severe visual impairment, the corrected distance visual acuity is between 0.1 and 0.05, or the visual field is less than 10%. In order to belong to ONCE in Spain, it is sufficient to meet only one of these requirements for severe visual impairment, either acuity or reduced visual field.

People with visual disabilities belonging to specific associations in the Basque Country and the Valencian Community have been invited to participate, where a large part of the members do not meet the requirements of legal blindness proposed by ONCE and therefore find fewer development opportunities. On the other hand, people affiliated with ONCE have also been invited. As part of the organization, they have their needs assessed regarding typhotechnical literacy and access to technical aids regarding the use of ICTs that are very specific and necessary for their development.

#### 2.3.1. Group 1

This group was made up of two people with severe visual impairment (less than 0.1 visual acuity and less than 10° visual field) affiliated with ONCE and four people with moderate visual impairment (less than 20° visual field) (Table 1). All the participants are residents of the Valencian Community and belong to a specific association for individuals with retinal pathologies.

#### 2.3.2. Group 2

As we can see in Table 2, this group was made up of six people with severe visual impairment (less than 0.1 visual acuity and less than 10° visual field) who reside in Aragón and are affiliated with ONCE, and two people with moderate visual impairment (less than 20° visual field) who belong to an association in the Basque Country.

## 3. Results and Discussion

When we talk about information and communication technologies, without a doubt, we have to say that they have brought about a revolution for the entire global population. If we talk about a more specific population group such as the one in this study, it is indeed a technological revolution that allows this visually impaired group to also participate in the advantages that this entails for everyone.

From this moment on, some differences should be marked among people with visual disabilities in Spain. On the one hand, people affiliated with ONCE, with legal blindness (blind individuals with low vision, severe visual impairment with less than 10° visual field), and on the other hand, the non-affiliated, where we find people with low vision, moderate visual impairment with less than 20° visual field.

We consider it necessary to understand the differences between people with legal blindness and the vast majority of people with low vision. As we have pointed out, both blind people and people with severe visual impairment can access ONCE affiliation if they also have Spanish nationality. However, people outside the health limits in terms of acuity and visual field suffer the consequences of helplessness regarding their digital literacy in the Spanish case. Even so, this is a worldwide benchmark for those who meet the minimum requirements because they can at least count on the help this organization offers its members.

### 3.1. Typhlotechnical, Computer, and Digital Literacy

Typhlotechnical, computer, and digital literacy is the capacity acquired by people with visual disabilities, which consists of having the necessary knowledge and skills in handling specific technical aids, whether typhotechnical or standard, which act as a link between these individuals and ICTs.

It also includes knowledge of the management and use of computer programs and applications that will allow these users to reach their educational goals, work or carry out activities necessary in daily life, such as: reading any text, press or publication, geolocation, online purchase of any kind, public or health administration procedures, access to social media, etc. At the same time, it also means accessing and interacting with all the information and content on the network that is of interest and necessary for these users.

In order to understand what this learning process represents, it is necessary to expose how these users begin acquiring knowledge to achieve that skill and thus take advantage of the opportunities offered by ICTs.

To begin with, users and experts agree on the need to accept their disability before a literacy process. We refer to the fact that each person, regardless of the pathology, in their process of visual loss, must assume this situation and everything that this acceptance implies. Thus, many of these users will require some time for psychological adjustment to the new situation of their visual state as there may have been a sudden or drastic accentuation of visual impairment. Sometimes the health diagnosis can be discouraging. According to one of the members of ONCE who is actively working (FG2.4), you are not as receptive when you do not accept your disability; they can talk to you about the smartphone, its adjustment and applications, but if you do not accept your disability and therefore your need, there is little that can be performed.

Affiliates have a specific program for access to ONCE. From the arrival of each case, there is a particular assessment by a multidisciplinary team in which the student goes through the different services that assess their needs. In the case of children, from the beginning of their training, they are always in coordination with the support teacher. This same process is repeated for adults and the elderly, an expert in typhlotechnology (EXP 1).

In addition, members perceive the need to learn and, thus, the support offered by the organization. The same perception is had from very early ages of affiliation. In the school stage, there have always been professionals from the organization determining their specific needs in terms of technical aids in order to cover them (magnifier, screen reader, braille display, etc.). These technical aids are provided to members to adapt to the study position.

“I stopped seeing when I was 13 years old. I had to get my Compulsory Secondary Education (ESO) and high school with a magnifying glass that I bought—making things very big and making a lot of tapes on history, economics, etc. I didn’t know there was another way to do it. I wrote without seeing what I wrote in exams until selectivity. Then I joined ONCE, and everything changed for the better. I would have really needed the help, but I didn’t know about it”. (Student, member, FG2.3).

“In 2014, at the age of 44, I was disabled from work. Then I began to use a magnifier, the Zoom text, and a year ago I started with a screen reviewer (Jaws).” (Disabled from work, affiliated, FG2.7).

It should be added that there were people with low vision who, despite being affiliated, did not recognize the need for learning. However, they changed their minds during the period of confinement due to COVID-19. Wanting to interact with other people marked a turning point, and, consequently, the need to access ICTs arose.

“My children helped me with the virtual meetings that I had during the pandemic, and now, for this meeting, they had everything ready for me to connect.” (Work disabled, member, FG1.4).

“The confinement forced us to get to know social media. The video calls were wonderful because we were able to celebrate birthdays and meet as a group. If it weren’t for the lockdown, I wouldn’t have learned how to use them.” (Work disabled, member FG1.5).

“During the pandemic, since we had so much time, we learned to use many mobile applications and to make videos, TikToks, edit them.” (Unemployed, unaffiliated FG1.2).

Once the initiation was over, difficulties also appeared for the entire group of visually-impaired people, regardless of their degree of visual impairment or whether or not they were affiliated with ONCE.

“I went to an ONCE course, but the truth is that I had to take the train by myself, and in the end, I only went for a couple of months. Then I stopped going because between the cane and the computer that I had to carry, everything was very complicated.” (Work disabled, member, FG1.4).

“I took a course at ONCE, quite superficial, and they installed the applications on my device. But the truth is that I ended up removing all of them because I didn’t know how to use them. To be able to do it, you need someone to teach you calmly. Learning to use them is not easy.” (Work-disabled, affiliated, FG1.5).

The range of causes that make this learning difficult includes reasons unrelated to the users and others of a personal nature to each individual. Concerning this topic and based on the fact that there are different technologies and degrees of visual impairment, the expert in typhlotechnology, EXP 1, has pointed out that literacy goes through different phases. While a sighted child can use a keyboard with only one finger, a child, in the case of severe visual impairment, must learn to use the keyboard for which typing is essential; then learn to use the screen reader with the respective key commands; and if necessary, also learn to use the braille line or typhlotechnical adaptation that is recommended for him.

This supposes the acquisition of the knowledge and the necessary practice to correctly use these specific technologies, as well as the management and use of programs and applications to finally access the information on the network. For this, knowledge of keyboard commands in the case of the computer is necessary, as well as finger gestures in touch technology used in smartphones.

The visually-impaired user’s objective is to know how to use technical aids correctly and navigate the Internet to reach the information point required by the user on the network. Likewise, users affiliated with ONCE share the same experience in terms of literacy (low vision expert EXP 3).

“The first thing they teach you is typing because to learn how to use a screen reviewer, you have to use keyboard commands, and you have to do this without seeing the keyboard.” (Female worker, affiliate, FG2.4).

At this point, the figure of the typhlotechnology and braille instructor (ITB) appears, who is the professional in charge of providing the necessary training to the user in the use of ICTs. Today, they play a fundamental role. For example, they teach the user to interact with a screen reader, not only on a computer but also on a smart television, multimedia devices, etc. They also teach how to use specific technology for the blind, such as the braille line or braille speak (EXP 2 accessibility expert).

The importance of these professionals is undeniable given the manifest need experienced by this group with visual impairment. Today, there is a need for more instructors in typhlotechnology, bearing in mind that there are autonomous communities in Spain, such as Navarra and La Rioja, that have only one instructor, Aragón two, Castilla la Mancha and the Basque Country three or Castilla León five, who must attend to all their respective provinces. Therefore, it is necessary to train adult students and, of course, older people since there are more and more things to learn regarding the use of computers, smartphones, tablets, and applications.

In the case of the elderly, there is an unequal use of mobile technologies [32] related to multiple factors: level of schooling, work experience, socioeconomic status, as well as insecurities and fears of lack of appropriate instruction regarding technology, etc. To this, the difficulties arising from the physical and cognitive deterioration typical of the age, such as loss of vision, fine motor skills, and memory, are added. These difficulties are accentuated by the lack of mobile devices and applications designed to respond to the particular characteristics of this group age. This suggests that as the expert in typhlotechnology affirms, the number of these professionals is insufficient to cover the affiliates’ demand, and to this, we must add that typhlotechnology and braille instructor profession is not studied anywhere other than ONCE (EXP 2 accessibility expert).

Outside ONCE, no professional field has the methodology, didactics, and computer knowledge to offer guidance and training to users outside ONCE. There is no professional with said qualification outside this organization. This shows the inequality present between users with blindness and low vision (severe visual impairment) who are affiliated with ONCE and other users with low vision (moderate visual impairment) who are not a part of once ONCE regarding the acquisition of information and training on the advantages of knowledge for the use of these technologies, as well as the recommended tools for access to ICTs.

“I would like to learn more; there are many things that can come in handy, but how do I know.” (female worker, unaffiliated, FG1.3).

“I don’t use accessibility technologies, but I’d like to know more because there may be something that could come in handy. I have assumed having to get very close to the screen and enlarge everything a lot.” (student, unaffiliated, FG 2.1).

As verified by the low vision expert (EXP 3), there is insufficient information. When a person is presented with needs associated with low vision, they do not know where to turn. In Spain, there is no place of reference where they obtain information and advice on existing technical aids and the corresponding literacy to use said aids in order to favor access to information for said users.

Likewise, users comment on the significant problem of not knowing everything that surrounds low vision as a disadvantaged group since there is a group of the population that has a low vision (moderate) and cannot access the ONCE because they do not meet the requirements (worker, member, FG2.6). This group is left unprotected in terms of the information it can access. To this, we must add the general misinformation, where no difference is made between visual impairment or whether or not they are affiliated with ONCE.

“In my case, I only use my magnifying glass with which I can read the newspaper, and that is enough for me. I miss the mobile phones from before, now I only have Whatsapp, and I don’t use it. I prefer to call and be called. I have had 200 messages on Whatsapp, but since it is difficult for me to read them, they stay there.” (disabled from work, member, FG1.4).

This shows a lack of information on where people with low vision can go for advice, as well as a lack of disclosure of the ways to access ICTs and the opportunities they offer to access and contribute to the information.

### 3.2. The Usability of ICTs by People with Visual Disabilities

When we talk about the usability of ICTs for people with visual disabilities, we are referring to the conditions that allow this group of people to use technological devices, programs, and computer applications, as well as the online services included in ICTs. Although for global users, these constraints can improve their user experience, for blind people and people with low vision, it can be the difference between accessing or not accessing these technologies.

For information and communication technologies to be usable, as low vision expert EXP 3 points out, it is first necessary to know if they are accessible. Let us take as an example a person in a wheelchair who wants to use a building: if that building is accessible, it will have a ramp, it will be usable, and the person in a wheelchair will be able to use the facility. The same happens with technologies: if they are accessible, blind people can use them to a greater extent, and in the case of people with low vision, if those technologies are accessible, they will be able to use them.

To begin with, we refer to web accessibility. Websites are made with many images that say nothing, with colors that cannot be changed, and where the text format is often not understood. There are guidelines when designing a website or mobile application so that it is accessible. For example, you have to describe images, forms, edit boxes, labeling, links, etc., and a screen reader has to be able to recognize all the elements on that page to say them (typhlotechnology expert EXP 1).

Experts and users agree on the need to comply with the recommended accessibility guidelines when creating and developing websites and mobile applications since aesthetics still prevail over functionality and not the other way around. On the other hand, we must not only think about people with visual disabilities but also about other groups (other disabilities, older people) since, when accessibility standards are not met, many people are left out of the information society (worker, member, FG2.6).

It is not about whether or not a website or mobile application is easy to use. It has to be accessible. It must meet the recommended guidelines that guarantee access to the entire global population. For example, buttons must be correctly labeled, all links, forms, and dropdowns must execute correctly, colors should be able to be inverted, the text format should follow recommendations, etc.

“The AAA (Royal Decree 1112/2018 (comply with the harmonized standard of the Accessibility Directive: EN 301549 v2.1.2:2018) collects criteria A and AA of WCAG 2.1) is often not present. The other day, carrying out a procedure in the public administration, I reached the digital signature after having filled everything out, and it was impossible for me to do so. I had to ask one of my children for help. After two hours of being there, he had achieved it in 30 seconds. It is impossible to use things that are not accessible.” (work disabled, member, FG2.7).

“In my bank, for example, you can’t enlarge the font, or invert the colors. I have told them, but they do nothing to change it.” (disabled from work, unaffiliated, GD2 8).

Accessibility does not harm anyone; on the contrary, it makes the user experience more comfortable for one hundred percent of users. We all (companies, institutions, etc.) must be participants in creating accessible products and services. Not from the point of view of social responsibility or sensitivity, which is fine, but it is not the objective. We are talking about access to information for all people as a universal right (accessibility expert EXP 2).

Everyone should be able to connect to the Internet and the rich complexity of knowledge it contains no matter where they are or where they are going. Mobile devices allow this to happen simply through mobile and wireless networks. However, from the point of view of hardware and software, the recipients’ unique characteristics are not always considered [33].

In this sense, the role of developers, computer scientists, and programmers is essential when creating products and services that meet the accessibility guidelines recognized in the world. To better illustrate what we mean, the accessibility expert EXP 2 clarifies that when we take a taxi, we deduce that the driver, in addition to driving, will know the traffic regulations and codes. Or at least he will know the vast majority of traffic regulations. You can find computer scientists who work for very powerful companies developing products and do not know the required accessibility standards, while that company is subject to a royal decree of which they are not aware. Failure to abide by these rules can lead to economic sanctions. Accessibility must stop being an accessory issue when creating and developing devices, products, and services. It must become a priority issue because, by developing accessibility in an already created product, it will be less functional and will entail additional financial costs.

Increasingly, the standard technology being developed allows the usability of technologies to the general population and people with disabilities. Although there will continue to be specific equipment for blind and visually impaired people, the trend is that a standard technology accessible to all can be used (exp1 typhlotechnology expert).

The screen reader is activated immediately when you buy a smartphone and remove its packaging. This phone is prepared for blind people; however, it is not a typhlotechnical device; we are talking about standard technology. Before, you had to buy a device and take it to ONCE to install the necessary programs (accessibility expert EXP 2). This denotes the opportunity for visually impaired users to indistinctly access computers or mobile phones.

“I use screen readers, above all, to read PDF documents and electronic books on mobile phones and computers” (female worker, unaffiliated, FG1.1).

“To work, I use the computer enlarger and invert the color contrast when I can” (unaffiliated worker, FG1.4).

“In order to study, I have the Windows magnifier, and I place the larger computer monitor” (unaffiliated student FG2.1).

For these purposes, accessibility is essential for a satisfactory user experience for everyone, including people with visual disabilities. The commitment of society must be firm, given that it is about the universal right to information for everyone. Nowadays, the appearance of standard technologies that increasingly favor the inclusion of people with visual disabilities is necessary. For this reason, the development and creation of products and services must meet all recommended accessibility guidelines.

### 3.3. Importance and Opportunities of ICTs for Blind and Visually-Impaired People

For all the participants in the discussion groups and experts, ICTs are very important today and have been a source of opportunities in many ways. For example, being able to communicate with anyone without the receiver knowing that you have a visual impairment—by email, for example—buying like any other person or having a friend online, has been a real discovery for people with visual disabilities.

“The population continues to believe that people who do not see do not communicate through social media or access the Internet, and that is not the case. Some students in their 90s, 80s and 70s know how to use technology and access, for example, Salud Informa (a Spanish public health application for managing appointments and vaccinations). Many sighted people of that age do not know, and our students do because necessity obliges. And it is even better in the case of touch technologies with mobile phones.” (typhlotechnology expert, EXP 1).

Although ICTs have a relevant importance for people with visual disabilities, it is necessary to point out what are the main opportunities denoted by users with blindness and low vision. As the FG2.6 worker, member, points out, in effect, since the Internet appeared and then mobile technology, it all changed, having instant access to information anywhere. You can read written information, recognize objects, access social media, information on the Internet, and GPS. The autonomy and independence it has meant for our group have been fundamental. You bring the cane and the GPS with you; if you are lost, you make a capture with a mobile application and know where you are (affiliated worker FG2.4).

ICTs provide a lot of security and, in my case, self-esteem, not feeling invalid or that you are outside of this society (student with low vision FG2.1). In addition, it promotes inclusion and that we can relate to all people (worker, affiliate FG2.5). When I got the disability, it seemed that the world was ending. However, when I discovered the possibilities that the smartphone and its applications gave me, I realized that I could continue doing things and interacting with people (unaffiliated and disabled from work FG2.8).

Thanks to technology, there are users with low vision who can work. Without its use, it would be something unthinkable or very complicated. Nowadays, I can work because I use screens; before, I used to do it on paper, but now it is challenging for me. There are times when I am afraid of what may happen in the future and whether I will have the capacity and the means to work in the long term since my vision will continue to degenerate (female worker FG1.3).

In the same way, being able to study or train for this group is feasible thanks to technology and typhlotechnical aids. “In my case, the greatest discovery has been the braille display. Had I known about it sooner, I would have saved myself a lot of time, paper, and transcripts because I needed to read to study and memorize, and the screen reader was not helpful for that (affiliated student FG2.2).

Being informed and participating actively has also been a point widely recognized by the users consulted. This is confirmed by the affiliated worker FG2.5, “reading a newspaper, accessing Twitter and participating like everyone else in the information society is very important to me”.

In addition to everything mentioned as benefits for this group, there is leisure, which adds personal autonomy. With the screen reader, you can read Whatsapp or follow series with audio descriptions (affiliated student FG2.3).

However, users recognize many difficulties outside the skills of handling and use of ICTs. Thus, as a worker affiliated with ONCE (FG2.4) comments, today and despite the fact that, in Spain, the laws require that accessibility be given with Royal Decree 1112/2018 “On the accessibility of websites and applications for mobile devices of the public sector”, we are very far from everything being accessible and the sanctions enacted by law being executed. As much as you have the configuration settings and know how to use those settings, if that content is not accessible, you can not do anything. Even if you know how to use ICTs and don’t want to be excluded, when it is not accessible, it disempowers you (affiliated worker, FG2.6).

It is also worth mentioning the marked digital gap that in this group is given both by age and by their visual disability. “If you add to this the existing misinformation about the great variety of mobile accessibility applications that could make our lives easier and that we are unaware of, everything is accentuated” (unemployed, unaffiliated, FG2.8). This is how misinformation about the use and existence of these technologies that can facilitate access to ICTs produces frustration and refusal to use them.

“You feel a lot of frustration when you want to do something and you can’t because you don’t know how.” (work disabled, member, FG1.5).

“There are times when you laugh that you don’t see something, but when there really is something that interests you and no matter how hard you try you can’t access it, it makes you very angry.” (disabled from work, member, FG1.4).

“As long as I have some residual vision, I’ll get by however I can, but when I lose it, I know it’s all over.” (disabled from work, unaffiliated, FG1.6).

## 4. Conclusions

ICTs offer a range of opportunities for autonomy, equity, inclusion, and participation in different areas: work, education, social, and personal, especially for users with visual disabilities [34]. This research reaffirms the primary need for accessibility for this group, both in technological products and services, for the usability of ICTs. Today it is found that many companies and institutions still do not recognize the importance of such accessibility and consider it, in many cases, as something accessory.

The need for typhotechnical, computer, and digital literacy for these users is also highlighted [35]. During the initiation phase, people with visual disabilities require the advice of qualified professionals [36] who assess their needs so that they subsequently obtain the necessary training to handle these technologies. These professionals, to this day, are still insufficient to provide this training in Spain.

In turn, the existing inequality between the opportunities of access to advice, training, and technical means for a large part of the people with low vision, who, in the Spanish case, are not affiliated to ONCE, is also highlighted. The irruption of technologies has surpassed everything imaginable for people with visual disabilities, which is why it is essential to make visible the great capacity [37] that ICTs have for these people. Indeed, they favor social participation [38] for this population group, practically on equal terms with the global population, since they allow real-time access to information, increasing their autonomy.

Experts and discussion groups underline the need for this accessibility [39], which is not being produced despite being recognized and collected in various academic articles as a primary characteristic of ICT usability and functionality. It is still not a generalized behavior by developers, computer scientists, and programmers, who continue, in many cases, to consider accessibility as a secondary issue when creating and developing products and services. However, access to information is a universal right of all people that also benefits the correct usability of the global population. Accessibility is not only about making it easy for people with functional diversity to access the network; it goes further. It allows access to a site to be the same for everyone, whether they have physical or mental problems, are elderly, have problems with the connection to the network or their device, or other external factors occur—such as lighting and noise that prevent a correct reading or interaction.

Currently, there are still companies and institutions for which, when developing their websites, mobile applications, or intranet, accessibility is only a matter of CSR policies or simply of image. However, it is not that they are designed only for people with disabilities but, in general, for any user who, for whatever reason, cannot browse said website.

We must reflect on the little importance given to the development and creation of accessible devices and services and the training given to professionals who develop these technology products and services in terms of current legislation. Web accessibility is still a pending issue to be dealt with extensively, both nationally and internationally. However, small glimmers of hope are beginning to be seen, encouraged by the institutions, as is the case of the Accessibility Standards of the Portal of the Electronic Public Administration of the Government of Spain [18], where the resources are grouped that allows specifying the characteristics that the contents available through Web technologies on the Internet, Intranets and other types computer networks must meet so that they can be used by most people.

On the other hand, this study highlights the need to disseminate the opportunities offered by ICTs to the entire group of people with visual disabilities. Many are unaware of its existence and, therefore, of everything that could contribute to improving and facilitating their autonomy.

ONCE in Spain is a special case that should serve as an example in other countries. It plays a highly relevant role for its members (Spaniards with legal blindness) as it provides them with access to ICTs, technical and typhotechnical aids, as well as the necessary literacy for the usability of said technologies. However, it must be emphasized that it leaves out a large number of people with visual disabilities who do not meet their requirements for access to the organization. This is, in itself, a costly problem to solve, but one in which the state should intervene.

In Spain, there are more than one million people with visual disabilities [4], including people with blindness and low vision. Within that amount, there are currently 70,462 people of Spanish nationality who are affiliated with ONCE and have recognized legal blindness [8], which represents only 7% of the total. These figures leave out 93% of people with visual disabilities, with hardly any alternatives to access an assessment of their condition, a tailor-made literacy process, and access to recommended technical aids.

Although there are specific associations for people with visual disabilities in Spain, we cannot forget that they are associations that do not have the resources, technical means, or qualified professionals to train in the use and exploitation of ICTs, in most cases.

Concerning the development of standard technologies that are usable by visually impaired people, these entail the creation of products and services that are no longer exclusively typlotechnical, favoring inclusion since both blind people and people with low vision can access content regardless of their blindness or visual impairment, as does the global population.

The following steps should address how to improve user experiences for people with low vision and thus favor the processes of inclusion, equity, and participation.

## Figures and Tables

**Table 1 ijerph-19-10782-t001:** Participants and characteristics of Focus Group 1.

Identifier	Characteristics
FG1.1	36 years old, female, worker, moderate visual impairment
FG1.2	34 years old, male, unemployed, moderate visual impairment
FG1.3	41 years old, female, worker, moderate visual impairment
FG1.4	44 years old, male, disabled from work, severely visually impaired, affiliated with ONCE
FG1.5	56 years old, female, disabled from work, severely visually impaired, affiliated with ONCE
FG1.6	60 years old, male, disabled from work, moderate visual impairment

**Table 2 ijerph-19-10782-t002:** Participants and characteristics of Focus Group 2.

Identifier	Characteristics
FD2.1	21 years old, male, student, moderate visual impairment
FD2.2	28 years old, female, student, severe visual impairment, affiliated with ONCE
FD2.3	30 years old, male student, severely visually impaired, affiliated with ONCE
FD2.4	35 years old, female, worker, severe visual impairment, affiliated with ONCE
FD2.5	45 years old, female, worker, severe visual impairment, affiliated with ONCE
FD2.6	48 years old, male, worker, severe visual impairment, affiliated with ONCE
FD2.7	52 years old, male, disabled from work, severely visually impaired, affiliated with ONCE
FD2.8	58 years old, woman, disabled from work, moderate visual impairment

## Data Availability

Data collected from participants are confidential.

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
