# Peer review of "The Usability of ICTs in People with Visual Disabilities: A Challenge in Spain"

_ijerph, 2022, doi:10.3390/ijerph191710782_

Round 1

Reviewer 1 Report

This article addresses information and communication technologies (ICTs) use by people with varying degrees of blindness. It is a qualitative case study based on the responses from a discussion of a panel of visually impaired people split into two focus groups and a separate group of experts with various competencies related to serving the visually impaired and blind. 

The authors provide highly descriptive accounts and highlights from the interviews with the focus groups and the experts. They address a wide range of topics, from pathology to psychology and technical literacy and skill. 

Overall, the article is well-written and properly organized. It is comprehensive in its treatment of the technology available to the visually impaired, and most importantly the shortcomings of such technologies. It addresses the fact that most software treat accessibility features as an accessory rather as a core function. 

The paper is specific to Spain and its ONCE organization, and presents it as a model to the world. As with anything, there are shortcomings: staffing, socioeconomic factors, and others. 

Overall, I think this paper merits acceptance and is an important contribution to research in software usability and accessibility. 

Author Response

Dear reviewer

We thank you deeply for your words regarding the research we have presented. It is a case study in Spain of how an institution like ONCE can serve as an example in other countries. Undoubtedly, it is only the first approach to the problems faced by visually impaired people and the use of new technologies.

Cordially

Reviewer 2 Report

This is a very interesting paper in which authors study the usability of ICTs with visual disabilities. Although the topic is interesting and falls within the scope of the journal, it still needs a little work to be done:

- Keywords: Translate them to English as in its present form they appear in Spanish.

- Introduction section: Is correct and contain a sum of the paper and an explanation of the dissabilities considered and what WHO said. Also ONCE organization is presented.

- Method: This section needs some work as I havne't found the sample and the characteristics of people. Please insert it. Are all of htem with the same % of visual dissability? Are there differences? The same occurs with experts, need some more definitions.

- Results, discussion and conclusions sections are correct.

Author Response

Dear Reviewer

We would like to thank you very much for the suggestions you have made to improve this article. In the attached document you can see how we have followed your suggestions. We hope they are sufficient.

We remain attentive
Cordially
